# Influence of Crack Width on Chloride Penetration in Concrete Subjected to Alternating Wetting–Drying Cycles

**DOI:** 10.3390/ma13173801

**Published:** 2020-08-28

**Authors:** Jun Lai, Jian Cai, Qing-Jun Chen, An He, Mu-Yang Wei

**Affiliations:** 1School of Civil Engineering and Transportation, South China University of Technology, Guangzhou 510640, China; 201510101120@mail.scut.edu.cn (J.L.); cvjcai@scut.edu.cn (J.C.); tellme_rose@163.com (M.-Y.W.); 2State Key Laboratory of Subtropical Building Science, South China University of Technology, Guangzhou 510640, China; 3School of Civil and Environmental Engineering, Nanyang Technological University, Singapore 639798, Singapore; he.an@ntu.edu.sg

**Keywords:** RC beams, transverse crack, chloride, wetting–drying cycle, corrosion

## Abstract

To investigate the durability of reinforced concrete (RC) beams under the combined actions of transverse cracks and corrosion, corrosion tests were conducted on a total of eight RC beams with different water–cement ratios and cracking states. The effects of the transverse crack width, water–cement ratio, and the length of the wetting–drying cycle on the distribution of the free chloride concentration, the cross-sectional loss of the tensile steel bars, and the chloride diffusion coefficient are analyzed. The results show that the widths of the transverse crack and the water–cement ratio of concrete greatly affected the chloride profile and content of the RC beam specimens. Specifically, the chloride contents in all the cracked RC beams at the depth of the steel bar exceeded the threshold value of 0.15%. As the width of the cracks increased, the chloride concentration and penetration of the cracked concrete beam increased. However, the chloride concentration at the reinforcement position did not seem to be obviously affected by increasing the wetting–drying cycles from 182 days to 364 days. Moreover, the decrease of the water–cement ratio effectively inhibited the penetration of chloride ions in the RC beam specimens. In terms of the cross-sectional loss of the steel bars, the average loss of the steel bar increases with increasing crack width for the beams with 182-day cycles, while the effect of crack width on the average loss is not as noticeable for the beams with 364-day cycles. Finally, a model is proposed to predict the relationship between the crack width influence coefficient, μ, and the crack width, w, and this model shows good agreement with the experimental results.

## 1. Introduction

Chloride-induced corrosion of a steel bar in concrete has become one of the leading causes of the deterioration of concrete reinforcement structures exposed to a saline environment [1]. Corrosion not only leads to the loss of the steel bar cross-section but may also decrease the bonding between the bar and the surrounding concrete, thereby promoting cracks and spalling of the concrete cover, deterioration of the anchoring strength between steel bars and concrete, and the eventual degradation of the structure loading capability and sustainability [2]. Among the numerous factors that affect the durability of reinforced concrete (RC) structures, chloride-induced corrosion is one of the main causes of steel bar corrosion, especially in structures exposed to seawater.

For RC structures exposed to seawater, four main environmental areas were divided according to the chloride ion content and degree of corrosion: the atmospheric, submerged, splash, and tidal zones. When an RC structure is immersed in seawater in the submerged zone, the risk of chloride-induced corrosion is very low due to the lack of oxygen and saturation. Nevertheless, in the other zones, the concrete structure is often subjected to seawater wetting–drying cycles, especially in the tidal and splash zones. Compared to the submerged zone, the chloride, water, and oxygen levels that exist in the tidal and splash zones are generally considered the optimum conditions for the initiation and propagation of concrete corrosion. Thus, these zones can be considered a significant factor in the design of structural durability. This has prompted a series of investigations relating to the permeability of chloride ingress into unsaturated concrete [3,4,5,6] since 2002, which has contributed significantly to the increase in knowledge of the mechanism of chloride transport in unsaturated concrete, especially under alternating wetting–drying conditions. In addition, there have also been several experimental studies about the effects of the water–cement ratio on the transport of unsaturated concrete subjected to wetting–drying cycles [7,8,9]. With the help of these experiments, the chloride erosion characteristics of uncracked concrete can be focused on by considering the real exposed conditions.

It is worth noting that the above studies focused on the permeability of chloride transport into uncracked concrete. However, RC structures under in situ conditions may suffer transverse cracks caused by external loads during their service life. Based on the current design code [10], lateral cracks within a specific width range are allowed. Commonly, the presence of these cracks has a noticeable influence on the penetration of chloride into concrete [11,12] and, as a result, accelerates the corrosion of steel bars [13,14].

Regarding the effects of transverse cracks on the durability degradation of RC beams, two issues have been the focus. The first is the penetration of chloride ingress into the transverse cracks of concrete. Aldea et al. [15] and Win et al. [12] reported that the permeability of chloride increases with an increase in crack widths ranging from 0.05 to 0.4 mm in concrete. Djerbi et al. [16] reported that the chloride diffusion coefficient increases with an increase in crack widths up to 0.25 mm. However, Rodriguez [17] and Jang et al. [18] found no trends of chloride penetration with various crack widths. Therefore, there are conflicting results regarding the effect of the transverse crack width on chloride penetration.

Steel bar corrosion in a cracked beam exposed to a chloride environment is regarded as the second issue. The presence of cracks has an obvious influence on the penetration of chloride into concrete [11]. However, more research is needed on how cracks have an impact on the corrosion process. The main issue relates to how cracks affect the corrosion process and whether this corrosion occurs in the long or short term. Some scholars [19,20] have previously reported that pathways for all aggressive media (O_2_,${\;\mathrm{Cl}}^{-}$, H_2_O, CO_2_, etc.) are created by cracks to accelerate corrosion initiation and penetrate the interface (slip and separate between the steel bar and concrete). However, it is still controversial whether the corrosion propagation period is affected by the crack width. Otieno et al. [21] reported that the presence of cracks increases the permeability of concrete, which causes accelerated chloride-induced corrosion; furthermore, the corrosion rate also increases as the crack width increases. Some additional studies [13,22] expressed the view that crack width does not affect the corrosion propagation period. It can be seen from Figure 1 that different crack widths, as well as different methods of corrosion evaluation, have been used by different studies. However, based on the existing data, it is difficult to establish a direct relationship between the effect of transverse crack width and the corrosion of the steel bar under different cracked concrete conditions.

In summary, the existing experimental studies focused primarily on the mechanism of chloride penetration into non-cracked concrete, while relevant researches into the chloride penetration and steel bars corrosion of cracked RC members exposed to chloride environments remain scarce and did not gain a clear understanding of the effects of the transverse crack width on the chloride transport and the steel bar corrosion propagation period. Therefore, an experimental program was conducted on a total of eight RC specimens exposed to a natural chloride environment with alternating wetting–drying cycles in the present paper, aiming at studying the effects of the transverse crack width, water–cement ratio, and the length of the wetting–drying cycle on the distribution of the free chloride concentration, the cross-sectional loss of the tensile steel bars, and the chloride diffusion coefficient. Finally, a model for predicting the influence of the transverse crack width on the chloride diffusion coefficient is presented.

## 2. Experimental Program

### 2.1. Test Specimens

This experimental study included eight RC beams in two series with dimensions of 150 × 200 × 1000 mm and with a concrete cover thickness of 25 mm. Figure 2 shows the arrangements of the longitudinal reinforcement and stirrup of the specimens. Specifically, hot-rolled HRB400 steel bars with the diameter of 14 mm are used for tensile reinforcements, while hot-rolled HRB 335 steel bars with the diameter of 8 mm are adopted for compressive bars and stirrup.

### 2.2. Material Properties

The test specimens were divided into two series: beam specimen series A (A1, A2) and beam specimen series B (B-01, B-02). The two series of beam specimens, A (A1, A2) and B (B-01, B-02), had different water–cement ratios but the same reinforcement arrangements. Table 1 provides the detailed concrete composition. Beam specimens A1 and A2 were designed with water–cement ratios of 0.3 and 0.35 respectively, and the B beams had a water–cement ratio of 0.4. The cement used for the two series of beams was a Grade 42.5 ordinary Portland cement (OPC). The maximum size of the rolled gravel was 15 mm. The fineness modulus of the sand, which was used as the fine aggregate, was 1.63. During fabricating each beam specimen series, three standard 150 mm cubes were also casted and cured together with the beam specimens under the same environment [10]. The 28-day compressive strengths of the concrete and the covariance (COV) are shown in Table 1.

### 2.3. Loading and Exposure Conditions

To investigate the combined actions of cracks and corrosion on a beam, after demolding and natural curing for up to 28 days, two series of beams were loaded together with a four-point loading system, as shown in Figure 3a. Thus, the crack would appear on the beam and last for its loading period. The maximum allowable crack widths set by various codes [10,23,24,25,26] are listed in Table 2. According to the Code for Design of Concrete Structures [10], this experiment was designed to produce three different crack widths (0, 0.1, and 0.2 mm). A load was applied by tightening the nuts on the thread, and the load value could be recorded through the load sensor presented in Figure 3a. The crack widths were observed in real time through the crack observation apparatus ZBL-F101 (Beijing ZBL Science & Technology Co., Ltd., Beijing, China), and the load value of the load sensor was recorded (values shown in Table 3). According to the different crack widths, there were three loading ratio levels, which refer to the ratio of the applied loads to the means of the calculated load limit from the flexural bearing capacity, for which the load ratio was 0%, 23.6%, and 36.7%. Figure 4 shows the specific location of cracks on the tensile surface. Since it was difficult to detect and correct the crack width in the wetting–drying cycle experiments, the effects of anchor relaxation creep were not considered. All crack widths used in this paper were the crack widths before exposure [27].

To ensure the one-dimensional penetration of the beams by chloride, after the loading process, all the concrete beam surfaces were covered with epoxy glue, except for the tensile surface of the concrete beams, to allow the ingress of chloride ions [27]. The beams were exposed to a 5% by weight NaCl solution by a brine pump, as shown in Figure 3b. The NaCl solution used in the experiments was produced according to the Chinese code (GBT 10125-2012) [28]. The resultant pH value of the solution is in the range from 6.0 to 7.0, and the content of other chemicals are almost negligible. Considering the proportion of the flooding tide and ebbing tide durations, they were placed in a wetting–drying cycle environment with 7 days of wetting followed by 7 days of drying, as shown in Figure 3c,d, and the temperature was the same as the natural environment in south China, with the monthly average temperature ranging from 17.8 to 26.4 °C.

### 2.4. Determination of Chloride Content

As shown in Figure 5, the specimens after the 182- and 364-day wetting–drying cycle tests were removed from the cistern and stored for 7 days before sampling to complete an entire wetting–drying cycle [27]. Once the 182- and 364-day wetting–drying cycle tests had been completed, the beams were cut to obtain concrete powder samples for the determination of chloride content. Sample positions on the uncracked beams were selected from the middle area of the beams, as shown in Figure 6a, and the sample positions of the cracked beams were selected from the middle area at each transverse cracked section of the beams, as shown in Figure 6b. Next, 50 mm diameter cores were cut from each specimen along the crack position and perpendicular to the test surface. All the cores were divided into 5 mm sections for the initial depth of 20 mm and then 10 mm sections from a depth of 20 mm to 50 mm. The core samples were then ground into powders per the Ministry of Communications of China standard JTJ 270-98 [29]. The water-soluble chloride concentration of the powdered samples obtained at different depths was also detected by the DY2501B-type chloride ion tester shown in Figure 7, which could obtain the value of free chloride content per unit mass of concrete at a given depth. Then, the chloride profile was plotted as a percentage of the concrete mass.

### 2.5. Cross-Sectional Loss

After the concrete powder samples was obtained for the determination of the chloride content test, the tensile steel bars embedded in the uncracked and cracked beams were extracted. A HCl solution with the specific gravity equal to 1.19 was used to remove the corrosion products of the naturally corroded tensile steel bars [29], as shown in Figure 8. The residual diameters of the corroded tensile steel bars were considerably different because of the varying shapes of pitting corrosion [27]; thus, measuring the diameter with a microcaliper was difficult. Accordingly, the steel bar weight loss was calculated as the result of a cross-sectional loss of reinforcement before and after corrosion. Depending on the corrosion pattern (more or less pitting corrosion) and corrosion conditions, the corroded reinforcement was cut into short pieces of different lengths. The length of the cut reinforcement depended on the corrosion details along with the reinforcement. The different lengths of the cut reinforcements were measured with Vernier calipers. The weighing accuracy of each cut section on the electronic balance was 0.01 g. The steel bar cross-sectional loss can be expressed as Equation (1). The original mass of the cutting sections was then calculated with Equation (2):(1)$$\Delta A_{\mathrm{av}}=\frac{m_{0}-m}{m_{0}}\cdot A_{\mathrm{av}}$$
(2)$$m_{0}=\rho \cdot A_{\mathrm{av}}\cdot L$$
where $\Delta A_{\mathrm{av}}$ is the average loss of the cross-section of the corroded reinforcements along the length of the cut section (unit: mm^2^), $A_{\mathrm{av}}$ is the nominal diameter of the reinforcements (unit: mm^2^), m and $m_{0}$ are the residual mass of the cut sections of the corroded reinforcement and nominal mass of the reinforcement (unit: g) respectively, ρ is the density of the reinforcements (unit: g/mm^3^), and L is the length of each cut section (unit: mm).

## 3. Experimental Results and Discussion

### 3.1. Chloride Profiles

Figure 9 provides the results of the free chloride content in all the cracked sections of the B beams. Figure 9a shows the chloride profiles of free chloride in the cracked sections of the beam specimen series B-01 after the 182-day wetting–drying cycles test, and Figure 9b shows the chloride profiles of free chloride in the cracked sections of the beam specimen series B-02 after the 364-day wetting–drying cycles test. Additionally, the mean values of the uncracked beam specimens B-01-1 and B-02-1 were plotted as red curves.

It can be concluded from Figure 9 that at the tensile reinforcement position (the blue line in Figure 9) for the uncracked Beam specimens B-01-1 and B-02-1, the chloride concentration was less than 0.15%. However, the chloride content at the same depth in all the other cracked beams exceeded 0.15% by mass of concrete. It is worth noting that the critical values of chloride concentration in RC structures vary considerably based on different researchers or standards. The threshold value of free chloride proposed by Federal Highway Administration (FHWA) [30] ranges from 0.15% to 0.18% by mass of concrete. This corrosion threshold value was confirmed by field studies of bridge decks [31,32], which showed that under some conditions, the water-soluble chloride content was as low as 0.15%.

Figure 9 shows that the width of transverse cracking applied to the concrete beam has a significant influence on the concentration of chloride. For the beam specimen series B-01 and B-02, with the increase in the transverse crack widths, the chloride content at a given depth was increased. This outcome of the differences in chloride profiles came from periodic changes in the porosity and the connectivity of the pores in concrete [33]. Under the bending moment, these changes in porosity and pore connectivity could inevitably lead to the initiation, growth, and linking of microcracks in the interfacial transition zone between the aggregates; additionally, the cement paste tended to be open and unstable, which would contribute to the increase in the connectivity of pores in the concrete. Therefore, the tensile stress caused by the bending moment increased the permeability of the free chloride, which would promote the ingress of free chloride into the concrete.

Comparing the free chloride profile curve of Figure 9a with that of Figure 9b, at the depth of the steel bar, the concentration of chloride ions when the crack width exceeded 0.2 mm was higher than that when the crack width was less than 0.2 mm. In addition, although the beam specimen series B-02 (364 days) suffered 182 more days of the wetting–drying cycles test than the beam specimen series B-01 (182 days), the chloride concentration at the reinforcement position did not seem obviously affected by that increased cycling.

### 3.2. Effect of the Water–Cement Ratio on Chloride Concentration

Table 1 lists the water–cement ratio observed for the specimens of each group. The water–cement ratios of the beam specimens A1 and A2 were 0.3 and 0.35, respectively. However, the water–cement ratio of the beam specimen series B-01 was 0.4. Figure 10 shows comparisons of chloride concentration of three water–cement ratio specimens.

The water–cement ratio and crack width both affect chloride transport in the cracked RC beams. Regarding the beams with three different water–cement ratios, it can be inferred from Figure 10 that the chloride concentration at a given depth (the reinforcement position in Figure 10) shows an increasing trend with an increase in the water–cement ratio, especially for the specimens with the high water–cement ratio. For instance, Figure 10d shows that the chloride concentration at the depth of the steel bar for the beam specimens A1, A2, and B-01-3 were respectively equal to 0.05%, 0.12%, and 0.29%, which meant that the water–cement ratio played a significant role in the penetration of chloride ions. As the water–cement ratio decreased, concrete could effectively inhibit the penetration of chloride ions

Figure 11 shows the free chloride concentration versus crack width at the steel bar positions of beams with different water–cement ratios. It can be seen from Figure 11 that the chloride concentrations of the A beams with a low water–cement ratio were all less than 0.15%. In the beam specimen series B-01 (182 days) with a high water–cement ratio, all the other positions were larger than 0.15%, except that the chloride concentrations in the no-crack sections were less than 0.15%. In the beam specimen series B-02 (364 days), the chloride concentrations in all parts were more than 0.15%. Furthermore, except for the beam specimen A1, whose water–cement ratio was lower than 0.35, the concentration of chloride ions appeared to approximately increase as the crack width increased.

### 3.3. Corrosion Maps

The corrosion maps of the beam specimens B-01-2 and B-02-3 after the 182- and 364-day wetting–drying cycles test respectively, are shown in Figure 12. Figure 12a shows the corrosion map of the tensile steel bars in the beam specimens B-01-2 after the 182-day wetting–drying cycles test and the transverse cracks caused by external forces. The zone in Figure 12 shaded in dark red represented pitting corrosion that was observed vertically from the tensile face direction, and the hatch-lined zone in Figure 12b represented general corrosion. After being cleaned with a HCl (the specific gravity equal to 1.19) solution to remove the corrosion products, it was found that all the corrosion shown in Figure 12a was pitting. When the pitting range around the crack increased, the pitting corrosion might expand to general corrosion (as shown in Figure 12b) over time.

The corrosion map of tensile steel bars in the beam specimen B-01-2 are shown in Figure 12a and demonstrate that after the 182-day wetting–drying cycles test, most of the pitting occurred near the transverse cracks. Due to the transverse cracks, chloride penetrated more easily into the concrete [27], so the chloride concentration at the crack increased accordingly. The chloride content of the steel bar passivation film quickly exceeded the threshold value, causing decomposition and initiating corrosion. This observation demonstrated that transverse cracks played an important factor at the beginning of corrosion [34]. The portions of the steel bars near transverse cracks acted as anodes, while the other parts acted as cathodes, causing macro-cell corrosion. All aggressive media (O_2_,${\;\mathrm{Cl}}^{-}$, H_2_O, CO_2_, etc.) promptly penetrated to the interface of the steel bar passivation film through the transverse cracks, and the chemical microenvironment changed. However, the size heterogeneity and nonuniformity of concrete inevitably made the chemical microenvironment of the inner steel/concrete interface nonuniform. As a result, the steel bars in concrete tended to have pitting corrosion in a local area at the early stage of the initiation and development of steel bar corrosion. Notably, the cracks might heal because of the accumulation of corrosion products; thus, the steel bars could be passivated at the crack tips due to the balance of the chloride concentration around the fracture path. Therefore, there was no apparent regularity between the transverse crack width and the initial positions of the corrosion, which meant that instead of the transverse crack width, the significant parameter of the corrosion of steel bars was the presence of cracks.

Figure 12b shows that the corrosion distribution is irregular, with a random distribution along the steel bar; therefore, the corrosion distribution was different for two different tensile steel bars. However, according to the findings in Section 3.3, chloride penetration in the cracked concrete was affected by cracks [35]. Thus, the presence of cracks led to an accelerated penetration of free chloride, and the internal microcracks caused corrosion to spread more easily along the steel bar. Once the reinforcement passivation film in the area located between the cracks was decomposed by chloride, new points of corrosion activity could be continuously formed.

### 3.4. Cross-Sectional Loss Distribution of Steel Bars

The tensile bars were extracted from the beams to observe the corrosion of the steel bars. The calculation method of the diameter loss of the corroded steel was introduced in Section 2.5. Figure 13 presents the cross-sectional loss at different positions along the steel bars of the beam specimen B-02-3 subjected to the 364-day wetting–drying cycle test. The distribution of corrosion on the steel bar was asymmetrical.

Several parts that were labeled with zone numbers in Figure 13 were divided depending on the transverse cracks of the beam specimen B-02-3. It is worth noting that the more substantial cross-sectional loss of the steel bar was located near the middle part of the beam, which was the corresponding maximum moment region. Furthermore, the maximum cross-sectional loss of the back-steel bar and the front-steel bar was located at the transverse crack between zones B and C and zones C and D, respectively. As described in the section corrosion map, the initial corrosion always occurred at the intersection of transverse cracks and reinforcements. Then, corrosion expanded quickly between the cracks.

To reveal the degree of the steel bar corrosion distribution more quantitatively, frequency histograms of the two wetting–drying cycle samples were plotted in Figure 14a,b for statistical analysis.

It can be found from Figure 14 that for the RC beams subjected to the 182-day wetting–drying cycles test, the average loss of the steel bar was 0.09% for B-01-1, 0.24% for B-01-2, and 0.32% for B-01-3. These values meant that the average loss of the steel bar increased with the crack width.

In contrast, despite different transverse crack widths, the average cross-sectional losses of the tensile bars were similar for all the beams subjected to the 364-day wetting–drying cycles test: 1.31% for B-02-1, 1.2% for B-02-2, and 1.33% for B-02-3. These average losses were much higher than the losses observed with the 182-day wetting–drying cycles test. Therefore, it could be found that the average cross-sectional loss of the steel bars in the specimens was independent of the transverse crack width (0~0.2 mm) when the specimens were subjected to the ingress of chloride under long-term wetting–drying cycles.

Thus, the chloride-induced cross-sectional loss of the steel bars was related to the crack width of the RC beam in the early stage of corrosion. However, as the wetting–drying cycles test progressed, the effect of crack width on the corrosion of the steel bars gradually decreased. Some research reported that cracks only induced the depassivation of the steel bars around the cracks and that the corrosion rate was not influenced primarily by the crack width when the crack width did not exceed 0.3 mm [16].

### 3.5. Effect of Crack Width on the Chloride Diffusion Coefficient

Jin [7] found that outside the depth range (5 mm) of the convection zone, the main mode of chloride penetration was diffusion. The chloride diffusion coefficient is a physical quantity describing chloride transportation in concrete and is used to reflect the diffusion rate of chloride [35]. The free chloride concentration, mainly by diffusion, measured at different depths is described by Fick’s second law [36]. The chloride diffusion coefficients of different crack widths were obtained by fitting the chloride diffusion coefficient at each measurement point based on the principle of the least squares method. Gowripalan et al. [11] considered the chloride transport law when bending cracked RC structures under wetting–drying cycles, which could be described by the chloride diffusion model. However, Lay et al. [37] found that the chloride diffusion model could be determined using Fick’s second law, as shown in Equation (3). The corresponding analytical solution of Equation (3) can be obtained and expressed in Equation (4) by using the error function.
(3)$$\frac{\partial C}{\partial t}=\frac{\partial }{\partial x}\left(D\frac{\partial C\left(x,t\right)}{\partial x}\right)$$
(4)$$C\left(x,t\right)=C_{s}\times \left[1-\mathrm{erf}\left(\frac{x}{2\sqrt{Dt}}\right)\right]$$
where *C* (*x*, *t*) represents the chloride concentration at depth x and at exposure time t, *C*_s_ is the chloride concentration at the surface of the concrete, *D* represents the chloride diffusion coefficient, and erf () denotes the statistical error function.

Based on Fick’s second law, the data fitting toolbox in MATLAB was used to fit and analyze the data in Figure 9a,b according to Formula (4). The chloride ion diffusion coefficients for the different crack widths could be obtained. The chloride diffusion coefficients of the B beams (182 and 364 days) are shown in Table 4. It can be seen from Table 4 that the distribution of the chloride content within the depth of the diffusion zone was in good agreement with Fick’s second law, and the fitted correlation coefficient, *R*^2^, was basically above 0.9, which meant that the distribution of chloride concentration conformed to the diffusion law. Table 4 also indicated that crack widths had a certain impact on chloride diffusion and could not be ignored; therefore, as the crack widths increased, the chloride diffusion coefficient increased.

Kwon et al. [38] considered that the crack width was the main parameter affecting the chloride diffusion coefficient of concrete. Additionally, they normalized the fitted chloride diffusion coefficients and defined the crack influence coefficient, μ = *D*_a_/*D*_0_, as a ratio, in which *D*_a_ and *D*_0_ represent the chloride diffusion coefficient with varying crack widths and without cracks, respectively.

The variation law of the crack width influence coefficient, *μ*, and the crack width, *w* (mm), obtained from the beam specimen series B-01 and B-02 was fitted, as shown in Figure 15. It should be noted that the A beams were not included for calculation because they did not have the *D*_0_ value for uncracked concrete. The fitting formula was calculated with Equation (5):(5)$$\mu =24.195w+1,R^{2}=0.9709$$

To validate the accuracy of the proposed fitting formula, the chloride influence coefficients, $\mu_{\mathrm{cal}},$ obtained from the proposed fitting formula were compared with the chloride influence coefficients from the experimental results of this research. It is shown in Table 4 and Figure 16 that the calculated values had good agreement with the experimental values, staying within a tolerance of less than 15% for most of the specimens.

However, due to the limited range of variables chosen for the specimens in the tests, it seemed reasonable to define the limits of the proposed formula. Equation (5) was applicable when the concrete crack width was lower than 0.4 mm. Thus, further studies and experimental work are required.

## 4. Conclusions

The influences of transverse cracks and the water–cement ratio on chloride transport in concrete in NaCl solutions under different conditions of wetting–drying cycles were experimentally investigated in this paper. The chloride penetration and nonuniform cross-sectional loss of steel bars with the influence of various transverse crack widths and wetting–drying cycles of the concrete specimens were presented and discussed.

The transverse crack width played a significant role in the chloride profile. Apart from the uncracked concrete specimens, the chloride contents in all the cracked concrete B beams at the depth of the steel bar exceeded the threshold value of 0.15% (by mass of concrete) proposed in the literature. As the width of the cracks increased, the chloride concentration and penetration of the cracked concrete beam increased. However, the chloride concentration at the reinforcement position did not seem to be obviously affected by increasing the wetting–drying cycles from 182 days to 364 days.

The water–cement ratio was also an important parameter affecting the chloride content in the concrete specimens. At the steel bar position, the chloride concentrations of the A beams with a lower water–cement ratio were all less than 0.15%. In the beam specimen series B-01 (182 days) with a high water–cement ratio, the chloride concentrations at most positions were larger than 0.15%. In the beam specimen series B-02 (364 days), the chloride concentrations were all larger than 0.15%.

Only pitting corrosion was observed in the beam specimen series B-01 (182 days), while both pitting and general corrosion were observed in the beam specimen series B-02 (364 days). The most substantial cross-sectional loss of the steel bar was located near the middle part of the beam, which was the corresponding maximum moment region. In the beam specimen series B-01 beams (182 days), the average loss of the steel bar increased with increasing crack width, while for the beam specimen series B-02 (364 days), the effect of crack width on the loss was not as visible.

A model that had good agreement with the experimental values, within a tolerance of less than 15% for most of the specimens, was proposed to predict the relationship between the crack width influence coefficient, *μ*, and the crack width, *w*.

## Figures and Tables

**Figure 1 materials-13-03801-f001:**
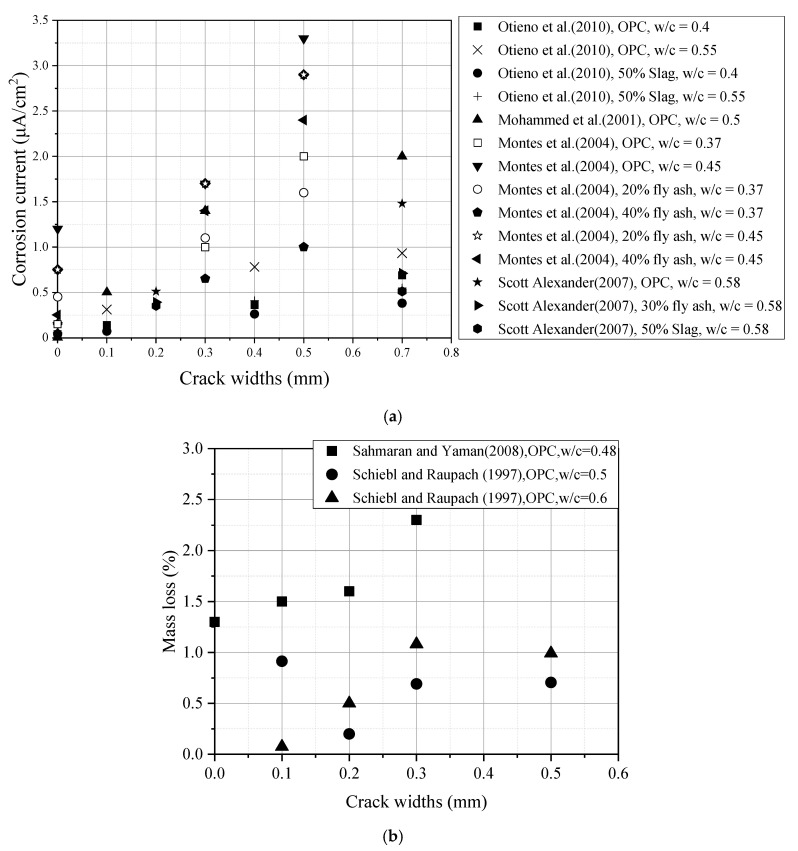
Effect of transvers crack widths on corrosion of steel bar in different concretes. (**a**) Corrosion current (OPC means the Ordinary Portland Cement); (**b**) Steel bar loss to corrosion (OPC means the Ordinary Portland Cement).

**Figure 2 materials-13-03801-f002:**
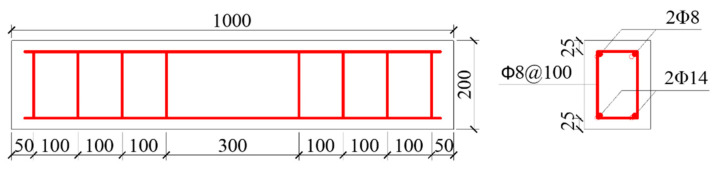
Layout of the reinforced concrete (RC) beams (all dimensions in mm).

**Figure 3 materials-13-03801-f003:**
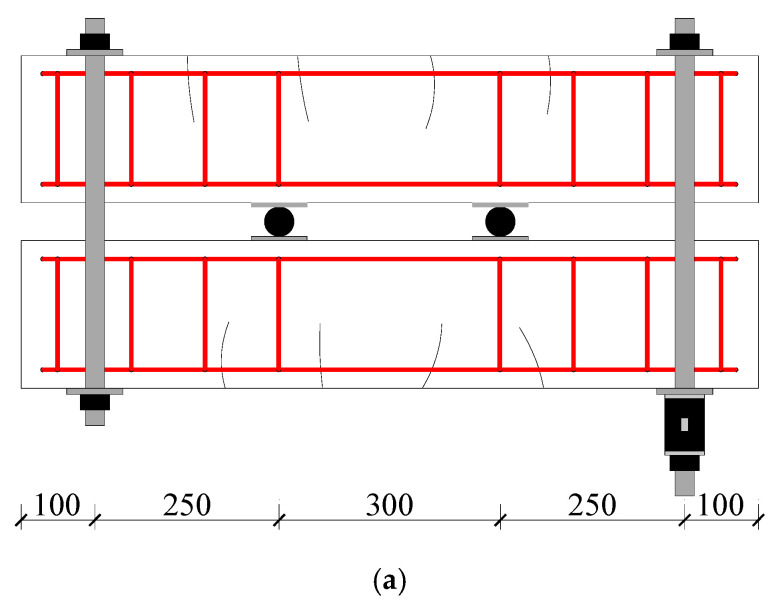
Loading system and exposure conditions. (**a**) Loading system (mm); (**b**) Exposure conditions; (**c**) Drying stage; (**d**) Wetting stage.

**Figure 4 materials-13-03801-f004:**
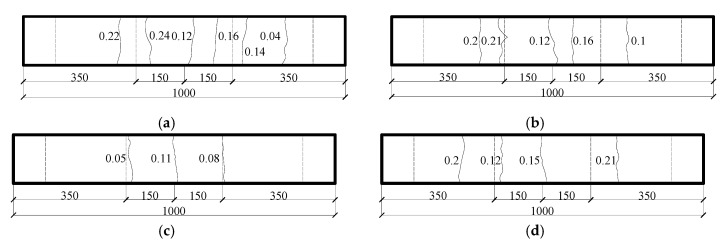
Specific location of cracks on the tensile surface (all crack widths used in this paper were the crack widths before exposure [27]) (**a**) A1 (mm); (**b**) A2 (mm); (**c**) B-01-2 (mm); (**d**) B-01-3 (mm); (**e**) B-02-2 (mm); (**f**) B-02-3 (mm).

**Figure 5 materials-13-03801-f005:**
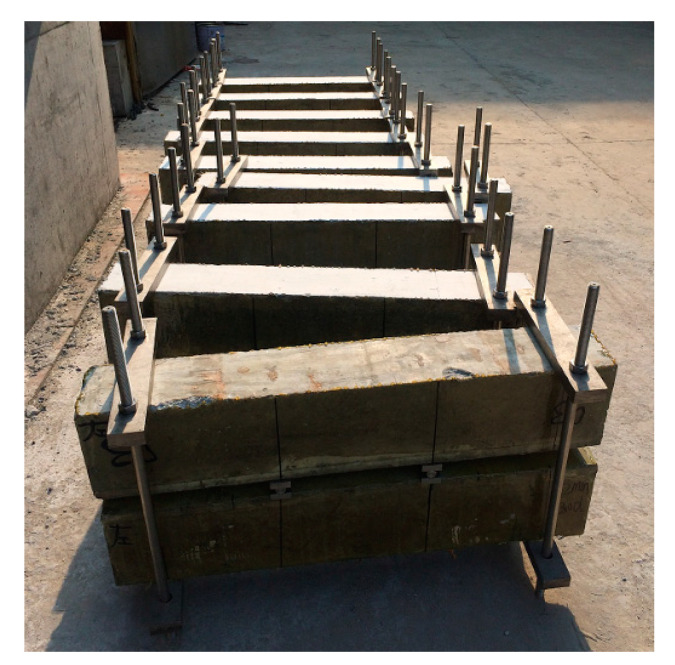
Specimens out of cistern.

**Figure 6 materials-13-03801-f006:**
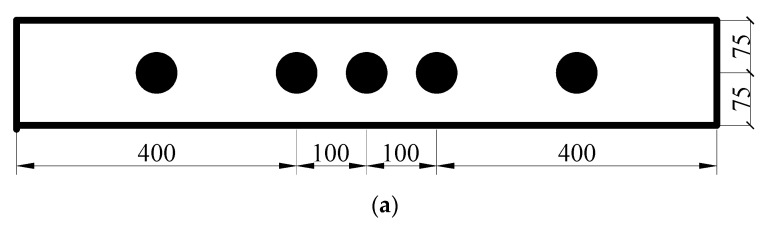
Schematic diagram of sample position for chloride profile (mm). (**a**) Uncracked beam; (**b**) Cracked beam.

**Figure 7 materials-13-03801-f007:**
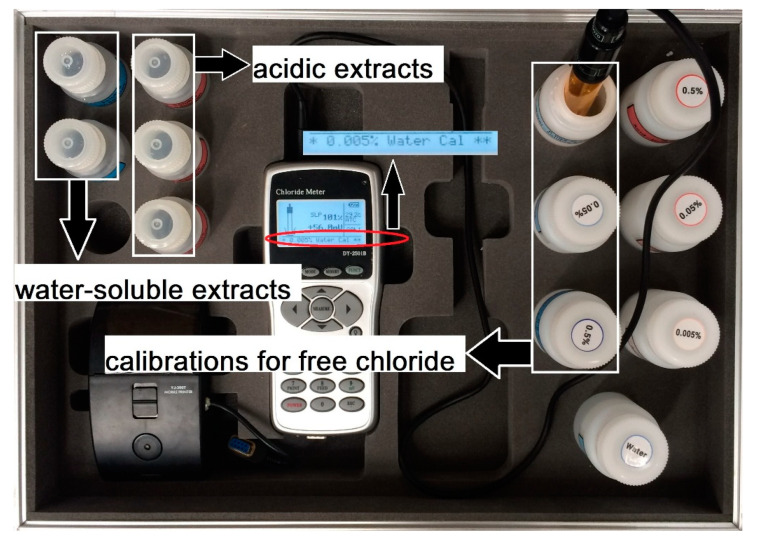
Determination of free chloride content.

**Figure 8 materials-13-03801-f008:**
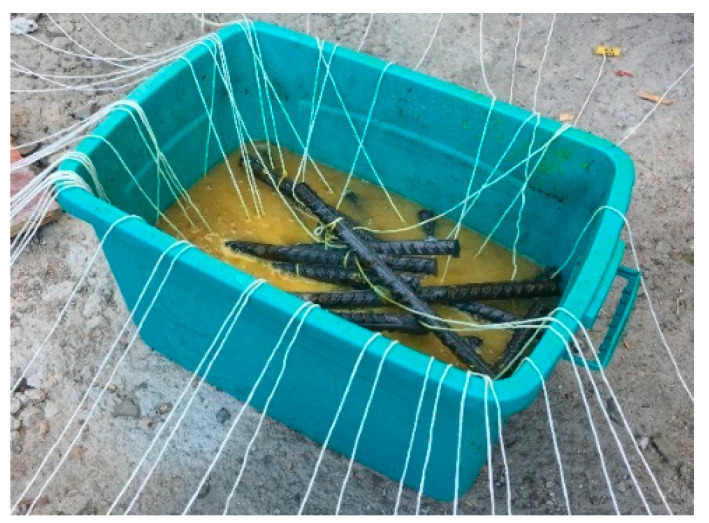
Cleaning with HCl solution.

**Figure 9 materials-13-03801-f009:**
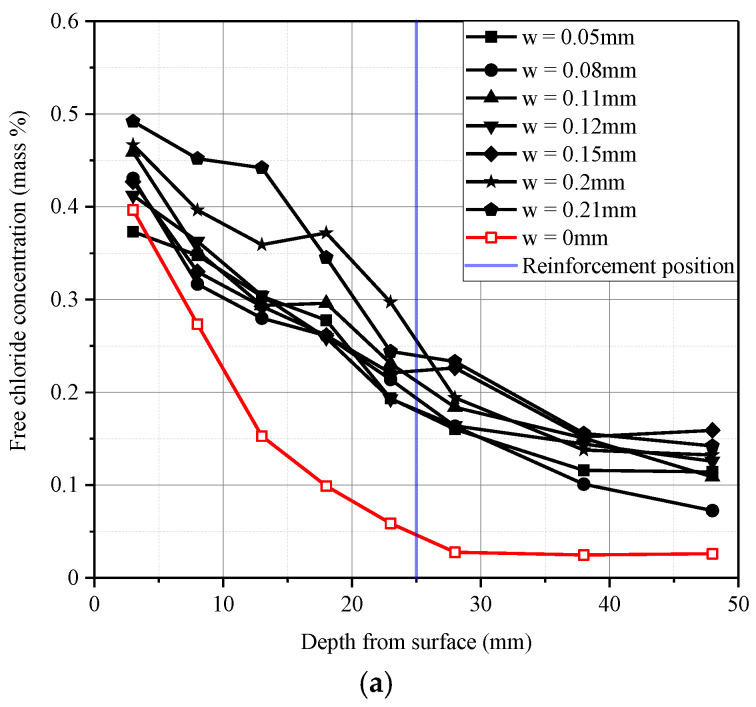
Free chloride profile for beam specimen series B-01 and B-02. (**a**) Beam specimen series B-01 (182 days); (**b**) Beam specimen series B-02 (364 days).

**Figure 10 materials-13-03801-f010:**
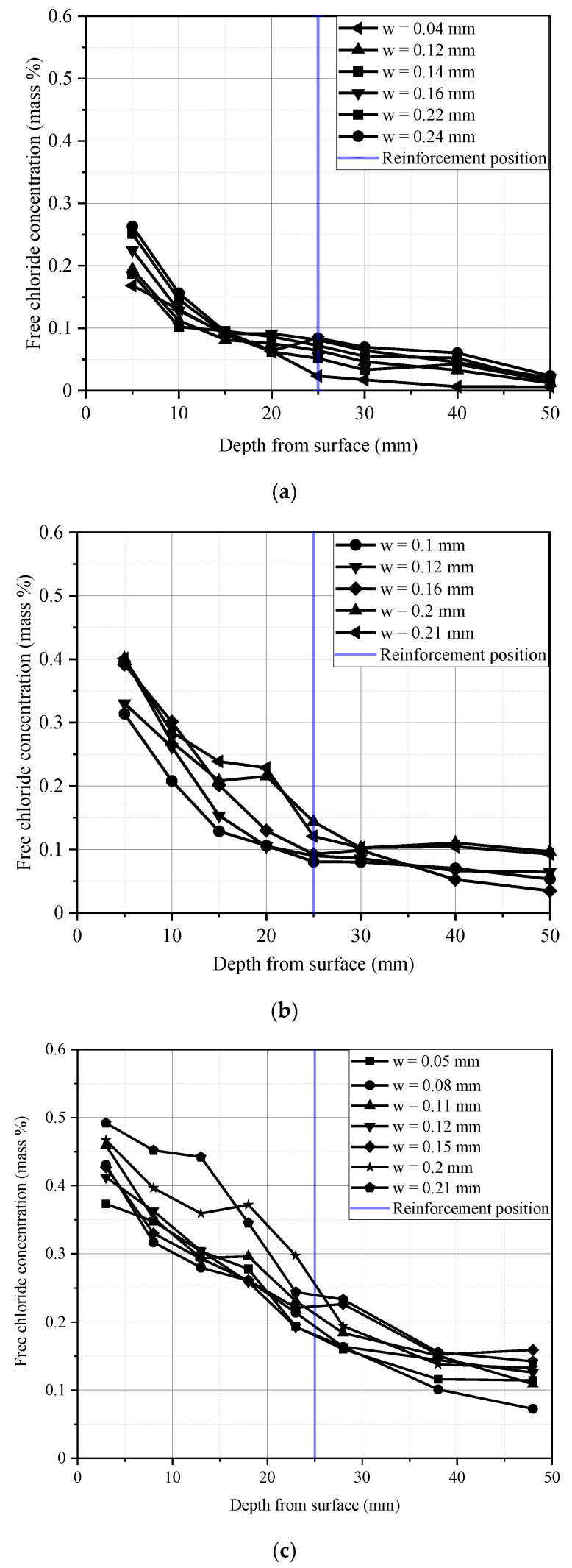
Free chloride profile for beam specimen series A and B after 182 days. (**a**) Beam specimen A1 (*W*/*C* = 0.3); (**b**) Beam specimen A2 (*W*/*C* = 0.35); (**c**) Beam specimen series B-01 (*W*/*C* = 0.4); (**d**) Comparison among specimens under different water–cement ratio.

**Figure 11 materials-13-03801-f011:**
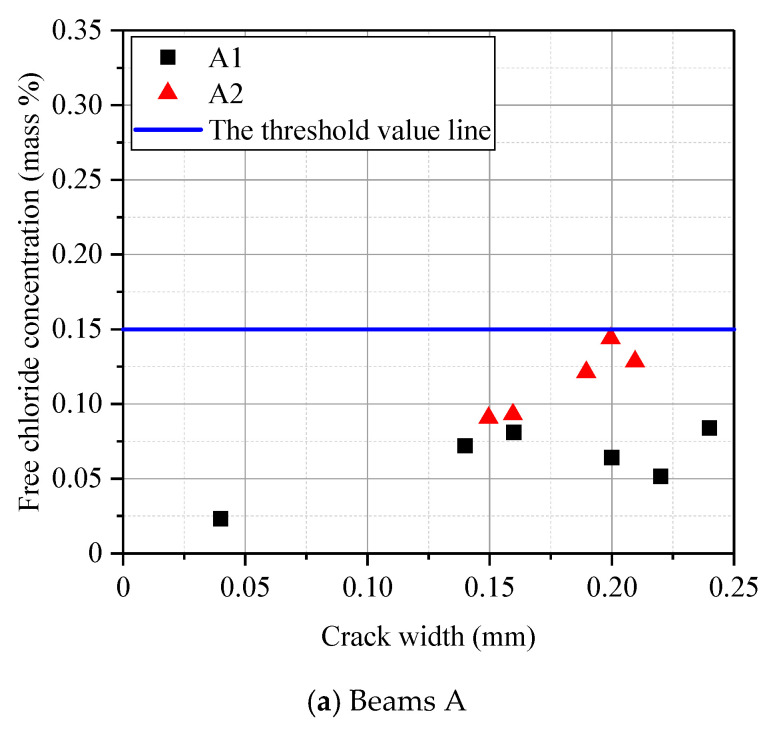
Free chloride profile at given depths (*x* = 25 mm) in cracked concrete section. (**a**) Beams A; (**b**) Beams specimen series B-01; (**c**) Beams specimen series B-02.

**Figure 12 materials-13-03801-f012:**
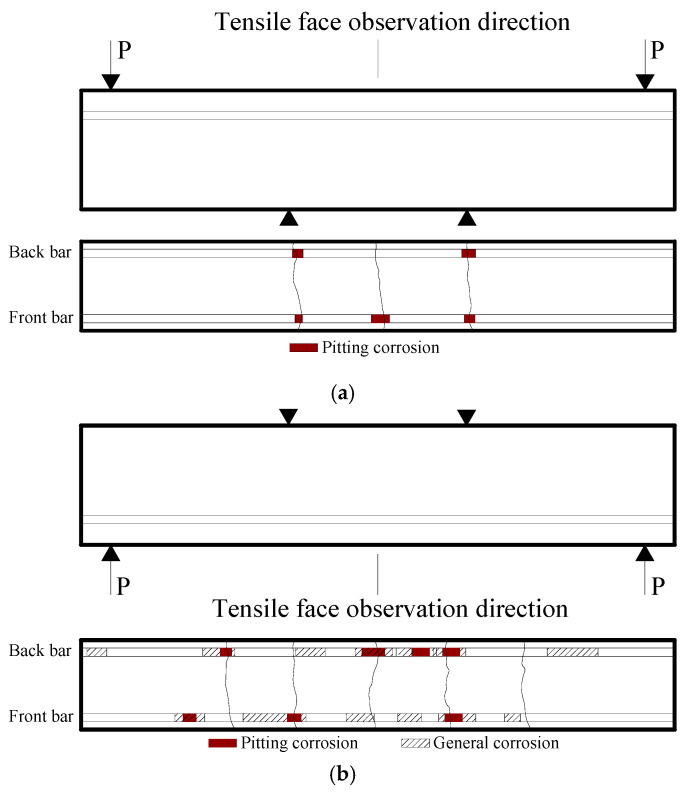
Corrosion map of tensile steel bars in beams. (**a**) B-01-2 (182 days; (**b**) B-02-3 (364 days)

**Figure 13 materials-13-03801-f013:**
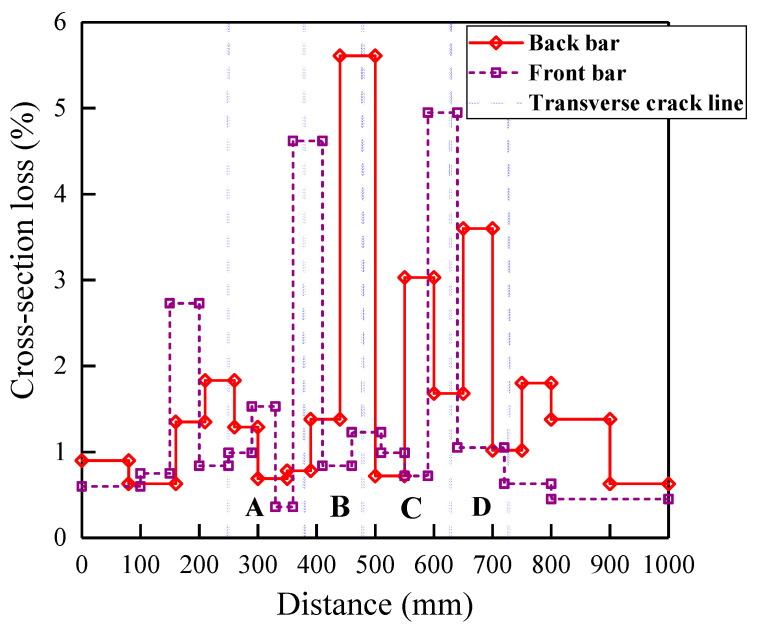
Cross-sectional loss of tensile bars in B-02-3.

**Figure 14 materials-13-03801-f014:**
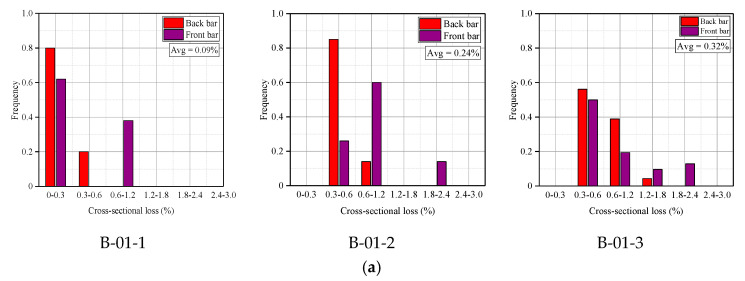
Frequency histogram of cross-sectional loss for corroded steel bars. (**a**) 182 days (Avg means the average value of cross-sectional loss for corroded steel bars); (**b**) 364 days (Avg means the average value of cross-sectional loss for corroded steel bars).

**Figure 15 materials-13-03801-f015:**
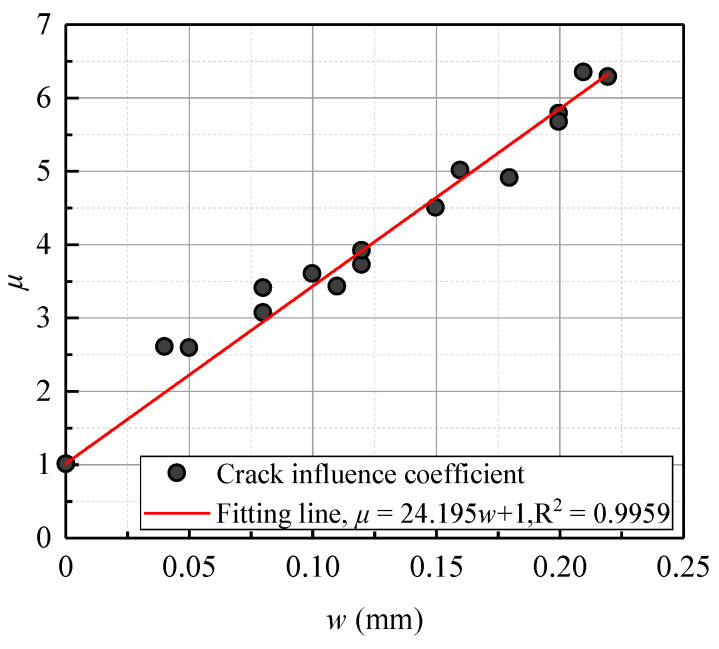
Fitting curve of crack width influence coefficient, *μ*, with crack width, *w* (mm).

**Figure 16 materials-13-03801-f016:**
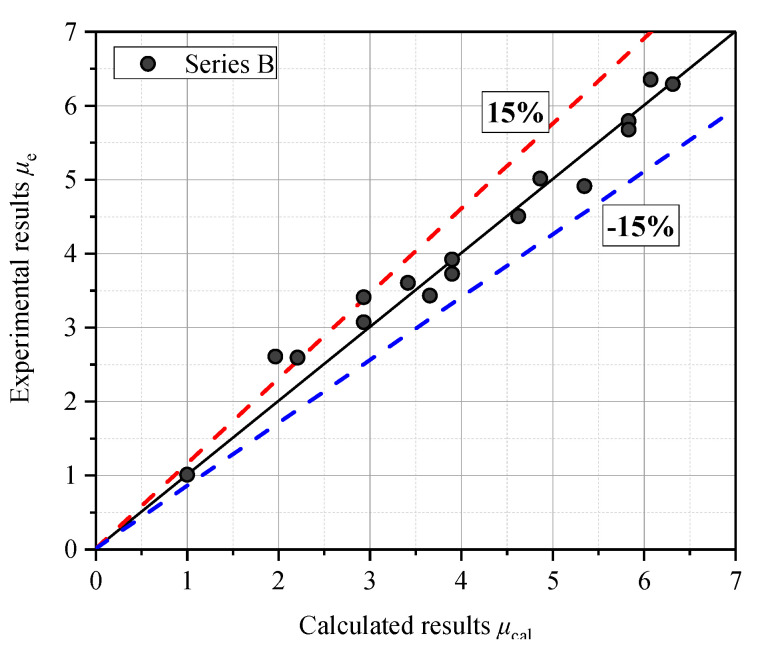
Comparison of experimental data and calculated results by Equation (5) in this research.

**Table 1 materials-13-03801-t001:** Concrete composition and strengths.

Specimen	*W*/*C*	Water (kg/m^3^)	Cement (kg/m^3^)	Sand (kg/m^3^)	Rolled Gravel (kg/m^3^)	28-Day Cube Compressive Strength (MPa)	COV
A1	0.3	182	606	1030	1988	54.9	0.041
A2	0.35	182	520	874	1695	52.0	0.037
B	0.4	182	455	755	1465	49.8	0.048

Note: *W*/*C* was the water-cement ratio; The size of sand was 0–5 mm.

**Table 2 materials-13-03801-t002:** Maximum allowable crack width set by various codes.

Code	Maximum Allowable Crack Width (mm)
ACI Manual [23]	0.15
CEB/FIP Model Code [24]	0.3
BS 8110 [25]	0.3
ENV 1998-1-1 [26]	0.3
Code for Design of Concrete Structures [10]	0.2

**Table 3 materials-13-03801-t003:** Load value of the beams.

Beam	*W*/*C*	Test Days (Days)	Load Value, *P**/P*_u_ (%)	Loading Mode
A1	0.3	182	37.6	Self-anchored
A2	0.35		37.6	Self-anchored
B-01-1	0.4	182	0	No-loading
B-01-2			23.6	Self-anchored
B-01-3			36.7	Self-anchored
B-02-1	0.4	364	0	No-loading
B-02-2			23.6	Self-anchored
B-02-3			36.7	Self-anchored

Note: (1) B-01/2-X means beam B, 01/2 means the first/second sampling test. X means beam number. (2) P means applied load, Pu means calculated limit load from flexural bearing capacity.

**Table 4 materials-13-03801-t004:** Fitting results of chloride diffusion coefficient.

Beam Series	*w* (mm)	*D* (×10^−12^ m^2^/s)	*C*_s_ (%)	*R* ^2^	*μ* _exp_	*μ* _cal_	*μ*_cal_/*μ*_exp_
B-01-1	0	0.4410	0.5338	0.9812	1.000	1.033	1.033
B-02-1	0	0.1427	0.5005	0.9692	1.000	1.016	1.016
B-01-2	0.05	1.1386	0.4254	0.964	2.587	2.282	0.882
	0.08	1.3501	0.4098	0.9743	3.068	3.031	0.988
	0.11	1.5086	0.4135	0.9552	3.428	3.781	1.103
B-01-3	0.12	1.6387	0.4439	0.9465	3.723	4.031	1.082
	0.15	1.9827	0.4166	0.9053	4.505	4.780	1.061
	0.2	2.5495	0.5183	0.9415	5.793	6.029	1.041
	0.21	2.7975	0.5662	0.9536	6.356	6.279	0.988
B-02-2	0.04	0.3715	0.4389	0.9811	2.603	2.000	0.768
	0.08	0.4861	0.4399	0.9433	3.406	2.983	0.876
	0.1	0.5138	0.4895	0.9812	3.601	3.475	0.965
B-02-3	0.12	0.5591	0.4344	0.9367	3.918	3.967	1.013
	0.16	0.7157	0.5032	0.9106	5.015	4.951	0.987
	0.18	0.7011	0.5397	0.9684	4.913	5.442	1.108
	0.2	0.8101	0.5061	0.9525	5.677	5.934	1.045
	0.22	0.8982	0.5843	0.9788	6.294	6.426	1.021

Note: ***w*** means the crack width; ***D*** means the chloride diffusion coefficient; *C*_s_ means the chloride concentration at the surface of the concrete; *R*^2^ means the correlation coefficient; *μ*_exp_ means experimental chloride influence coefficient; *μ*_cal_ means calculated chloride influence coefficient.

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
