# Peer review of "Influence of Crack Width on Chloride Penetration in Concrete Subjected to Alternating Wetting–Drying Cycles"

_materials, 2020, doi:10.3390/ma13173801_

Round 1
Reviewer 1 Report
Although the paper does not add any unexpected results but it is generally useful for RC structures exposed to seawater.
What is the main question addressed by the research?
The script tried to investigate the long term effect of crack width on concrete potintially affected by water containg chloride e.g. seawater.Is it relevant and interesting?
It is relevant and can be interesting to your journal readers.
I would suggest to expand more on the relation of this work with industrial codes used for concrete structures. Also please clarify the pH and other chemicals in the water used in the experiments.
Author Response
The authors would like to thank the reviewer for this comment. The response to the reviewer for the comment has been upload by Word.

Reviewer 2 Report
- The abstract of the article should include more detailed research results, according to the conclusions, as well as the globalism of these issues.
- Please correct the description on Figure 2.: "Φ8@100"
- The lines 240-246 describe the reasons for differences in chloride profiles. Have any tests been carried out on the porosity of concrete in the specimens taken? Have tests been carried out on SEM, concerning possible phase changes of the concrete microstructure? Then the causes of these phenomena would be more reliable.
- 10 (d) is interpreted in lines 270-271. Quoting: „For instance, Fig. 10(d) shows that the chloride concentration in the A1beam at the depth 271 of the steel bar was 0.15% by mass of concrete, A2 was approximately 0.15%,…”. But I see something else, am I wrong?
Author Response

(The authors gave the same response as above.)

Reviewer 3 Report
The paper studies the issue of influence of crack width on chloride penetration in concrete subjected to alternating wetting-drying cycles. The subject of the paper is very current and relevant. The effects of the transverse crack width, water-cement ratio, and the length of the wetting-drying cycle on the distribution of the free chloride concentration and the cross-sectional loss of the tensile steel bar as well as the influence of the transverse crack width and the length of the wetting-drying cycle on the chloride diffusion coefficient are analyzed.
There are some issues that require the corrections. Please see remarks below.
1.In the introduction the authors had mentioned some related papers in the literature. Could you please provide more comments to show the advantage and progress of the present researches in this study over previous works. What is the novelty of the paper?
2.Please correct the title of the figure 1, as from the description seems to analyze the effect of transverse crack widths of steel bar.
3.Please explain the abbreviation “OPC” in figure 1 / Ordinary Portland Cement /.
4.Please give the reference in the text to figure 4 .
5.Page 8, please explain “(sp gr 1.19)”
Author Response

(The authors gave the same response as above.)

Reviewer 4 Report
Generally, the submitted manuscript is interesting and well organized. However, its novelty is not clearly given, which should be highlighted in its revised version. Moreover, I have following comments that should be considered before paper acceptation:
- i) What was the steel type used for reinforcing bars?
- ii) In measurement the compressive strength, the standard used should be introduced together with information on measurement uncertainty and information on samples curing.
iii) How was the composition of concrete prisms designed? It would be much more interesting to apply some cementitious or other fine mineral admixtures in composition of tested materials. Please, comment.
- iv) The application of the Fick's second law for the description of chloride diffusion is far from reality and really too simple for actual research. It is well known, the chloride diffusion coefficient is highly dependent on chloride concentration, which was neglected. Please, use different model for calculation chloride diffusion coefficient. It is really necessary. See and refer following papers, where such methods are used.
Computational modelling of coupled water and salt transport in porous materials using diffusion advection model
By: Pavlik, Z.; Fiala, L.; Madera, J.; et al.
Conference: 3rd International Conference on Modeling, Simulation, and Applied Optimization (ICMSAO-09) Location: Amer Univ Sharjah, Sharjah, U ARAB EMIRATES Date: JAN 20-22, 2009
JOURNAL OF THE FRANKLIN INSTITUTE-ENGINEERING AND APPLIED MATHEMATICS Volume: 348 Issue: 7 Special Issue: SI Pages: 1574-1587 Published: SEP 2011
Experimental analysis of coupled water and chloride transport in cement mortar
By: Cerny, R; Pavlik, Z; Rovnanikova, P
CEMENT & CONCRETE COMPOSITES Volume: 26 Issue: 6 Pages: 705-715 Published: AUG 2004
- v) It is not clear how was the free chloride content measured and how it was calculated from the total salt concentration which involves also bounded chloride ions? Please, explain.
- vi) Do you have any idea about the content of bounded chlorides? Measurement of chloride binding isotherms might be done.
As the major modifications of the papers will be performed, I will recommend paper for publication as it aims at really crucial durability issue of concrete products.
Author Response

(The authors gave the same response as above.)

Round 2
Reviewer 4 Report
The authors addressed most of my comments in revised paper.